# Imperfect two-dimensional topological insulator field-effect transistors

William G. Vandenberghe[1] & Massimo V. Fischetti[1]

To overcome the challenge of using two-dimensional materials for nanoelectronic devices, we propose two-dimensional topological insulator field-effect transistors that switch based on the modulation of scattering. We model transistors made of two-dimensional topological insulator ribbons accounting for scattering with phonons and imperfections. In the on-state, the Fermi level lies in the bulk bandgap and the electrons travel ballistically through the topologically protected edge states even in the presence of imperfections. In the off-state the Fermi level moves into the bandgap and electrons suffer from severe back-scattering. An off-current more than two-orders below the on-current is demonstrated and a high on-current is maintained even in the presence of imperfections. At low drain-source bias, the output characteristics are like those of conventional field-effect transistors, at large drain-source bias negative differential resistance is revealed. Complementary n- and p-type devices can be made enabling high-performance and low-power electronic circuits using imperfect two-dimensional topological insulators.

[1] Department of Materials Science and Engineering, University of Texas at Dallas, 800W Campbell Road, Richardson, Texas 75080, USA. Correspondence and requests for materials should be addressed to W.G.V. (email: william.vandenberghe@utdallas.edu).

To obtain the best possible electrostatic control in electronic devices such as field-effect transistors (FETs), two-dimensional (2D) materials have to be used[1]. However, in practice this is proving to be very challenging. A first big challenge originates from the need to discover 2D materials with electronic-transport properties that exceed significantly those exhibited by silicon technology. A second challenge consists in devising an appropriate switching mechanism enabling the exploitation of the transport properties. A third, more practical, challenge is to find materials that can be grown with high quality and uniformity to enable the manufacturing of reproducible devices on a large scale. Efforts have been made to use 2D materials for conventional FETs[2,3] as well as alternative electronic devices operating based on tunnelling[4–6], ferroelectrics[7], spin, exciton condensates[8], phase-transitions[9,10] but, as of present no avenue has been found to overcome all of the aforementioned challenges of 2D materials.

Graphene, as an atomically thin material, exhibits a very high mobility but, unfortunately for FETs, it has no bandgap and no good alternative switching mechanism has been devised[11,12]. Opening a bandgap by using graphene nanoribbons drastically reduces the mobility[13,14] and the large sensitivity of the bandgap to the ribbon width makes graphene nanoribbons extremely sensitive to any edge roughness[14]. Exciton-based graphene devices[8] are likely to only work at low temperatures[15,16], tunnelling devices are expected to result in low drive currents[5,17,18], graphene devices based on transmission have low on/off ratios[12] and will inevitably suffer from imperfections introduced during the fabrication process.

The exploration of new 2D materials, such as transistion metal dichalcogenides, has shown some promise but mobilities are quite low[19] and defect levels in the materials are very high. Present material quality suffers from up to 7 orders of magnitude more defects[20] compared with industrial silicon impurity standards. Other more exotic 2D materials are found in phosphorene[2] (monolayer phosphorous in a puckered configuration) which was initially predicted to have a very high mobility[21] but more rigorous calculations reveal a much less exciting phonon-limited mobility of $\approx 200\,\mathrm{cm^2 V^{-1} s^{-1}}$ (ref. 22). Silicene[23,24] (silicon in a buckled hexagonal monolayer configuration) was similarly initially predicted to have a mobility similar to graphene[25] but properly accounting for scattering with flexural modes in the absence of horizontal mirror symmetry reveals a silicene mobility essentially zero for practical purposes[26].

As continuing research has moved towards heavier elements, the effects of spin–orbit coupling have become more important. In graphene, for example, the effect of spin–orbit coupling can be safely ignored[27] whereas in stanene[28,29] (tin in a hexagonal monolayer configuration), spin–orbit coupling opens a bandgap of 0.17 eV which is much larger than the thermal energy at room temperature. Particularly some of these materials like stanene, functionalized stanene, transistion metal dichalcogenides in the distorted tetragonal phase[30], $ZrTe_5$ (refs 31,32), bismuthene[33] and several other proposed materials, are 2D topological insulators (TIs)[34,35]. The TI nature guarantees the presence of edge states in 2D TI ribbons with excellent transport properties even at very large levels of material imperfections such as vacancies, doping or impurities. Proposals to make FETs by switching from 2D trivial insulators to TIs have been made since their inception[30,36,37] but unfortunately operating in this way requires unrealistically large electric fields (for example, $30\,\mathrm{MV\,cm^{-1}}$ in ref. 30). Three-dimensional (3D) TIs also have surface states whose presence is protected against imperfections. However, for transistor applications 3D TIs have severe disadvantages: a 3D TI will inevitably suffer from shunting paths through the bulk and through surfaces other than the surface on which the device is fabricated; the surface states of 3D TIs are effectively metallic making it hard to significantly move the Fermi level; and while the 3D TI surface states are also spin-polarized, making them possible candidates for spin-based memory devices, conduction is not ballistic in 3D surface states.

In this paper we study theoretically the electronic properties of TI FETs[38] whose operating principle is based on the promotion of back-scattering. We analyse the device performance by numerically solving the Boltzmann equation coupled with the Poisson equation. We account for intra-edge scattering due to phonons[39] and lattice imperfections such as edge roughness or defects. Using the Boltzmann equation ensures that Pauli's exclusion principle and the ballistic limit are respected. Modulation of the gate bias modifies the scattering strength in the device and we find that scattering with imperfections is beneficial for the efficient operation of the TI FET. We compare the TI FET with other devices in terms of elementary circuit performance and show that it is competitive with high-performance complementary metal-oxide-semiconductor (MOS) technology in terms of speed and competitive with other proposed energy-efficient devices in terms of energy consumption. We conclude that the TI FET can provide a high-performance low-power FET device without requiring defect-free materials.

## Results

**Edge states.** Figure 1a shows the band structure of a 2D TI as calculated using the Bernevig–Hughes–Zhang (BHZ) Hamiltonian[35] $H^{\mathrm{BHZ}}(K)$. Solving the Schrödinger equation yields

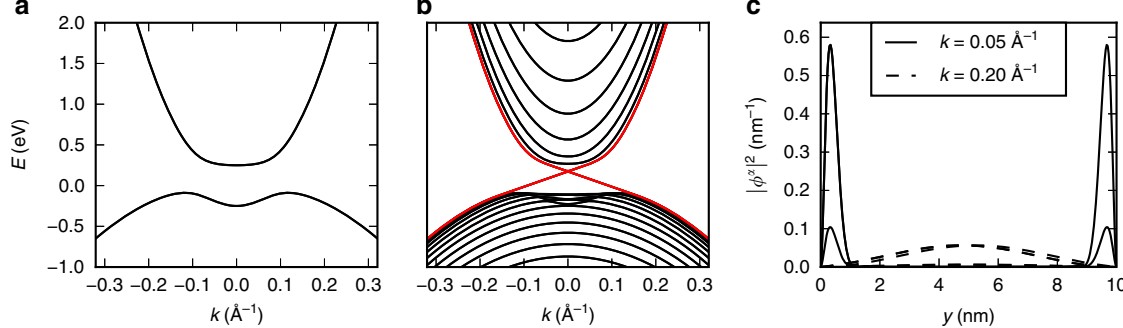

**Figure 1 | Topological insulator band structure and wavefunctions in bulk and ribbon.** (**a**) Bulk topological insulator band structure, (**b**) 15 nm topological insulator ribbon band structure and (**c**) the magnitude of the four wavefunction components of the valence band edge states for $k = 0.05\,\text{Å}^{-1}$ (solid) and $k = 0.2\,\text{Å}^{-1}$ (dashed). The states traversing the bulk bandgap in the ribbon band structure (indicated in red in **b**) are the topologically protected spin-polarized edge states. The states for $k = 0.05\,\text{Å}^{-1}$ lie in the bulk bandgap, are localized on the left and right edge and decay exponentially between both edges. The states $k = 0.2\,\text{Å}^{-1}$ (dashed line in **c**) do not decay exponentially and have a significant overlap.

the energies $E_j(k)$ and the ribbon wavefunctions $\phi_{kj}^{\alpha}(y)$, where $k$ denotes the momentum along the ribbon transport direction and $\alpha$ is an index running over the four degrees of freedom of the BHZ Hamiltonian. The BHZ parameters are chosen to obtain a band structure similar to the one of functionalized monolayer tin as determined from first principles[28,40], as discussed in the Methods section. Figure 1b shows the band structure of a TI ribbon with a width $w = 15$ nm. The ribbon band structure reveals the topologically protected edge states compared with the bulk band structure in Fig. 1a. Each band is twofold degenerate but for the edge states traversing the bandgap, the wavefunctions are localized on opposite edges, as revealed in Fig. 1c. The edge states have an almost linear dispersion in the bandgap with a high Fermi velocity ($5 \times 10^5$ ms$^{-1}$).

Edge states with opposite momentum $(k \to -k)$ on the same edge have opposite spin-polarization $(\uparrow \to \downarrow)$ because of time-reversal symmetry. The opposite spin-polarization ensures that phonon, edge roughness, defect and impurity intra-edge back-scattering is prohibited: the intra-edge matrix elements vanish. While for inelastic processes non-vanishing intra-edge matrix elements are possible[41], they are negligible for our purpose. The matrix element with edge states on the opposite edge does not vanish but for 'wide' ribbons, the matrix element is small and back-scattering is strongly suppressed. These arguments hold true for edge states with an energy in the bulk bandgap whose wavefunctions exhibit exponential decay away from the edge. On the contrary, wavefunctions associated with edge states whose energy is not in the bandgap do not decay exponentially and these states may have significant overlap as illustrated in Fig. 1c.

**Device structure and simulation**. We show an illustration of a TI FET and its working principle in Fig. 2. Specifically, we simulate a TI FET with a gate length $L_{gate} = 10$ nm and a ribbon width $w = 15$ nm. We solve the Boltzmann transport equation self-consistently with the Poisson equation in a region of length $L = 30$ nm and a channel and oxide thickness of 1 nm. We account for scattering with phonons whose spectrum, polarization vectors and deformation potentials are determined from first principles[42]. We also account for scattering with imperfections (having line-edge roughness (LER) in mind as a specific example), with strength measured by a parameter $U$. Additional details about the simulation method are given in the Methods section.

In Fig. 3a, the resulting distribution function of the conduction band is illustrated for a gate-source bias $V_{gs} = 0.1$ V, a drain-source bias $V_{ds} = 0.1$ V, and with significant scattering with imperfections ($U = 16$ eV nm). Because of the applied drain-source bias, the Boltzmann distribution is asymmetric with respect to momentum and current flows through the device. In Fig. 3b the distribution function for $V_{gs} = 0.5$ V is illustrated.

The distribution function and the associated charge density increase in the gate region as a result of the electric field induced by the gate and the resulting acceleration of the carriers in the source (0–10 nm) and drain region (20–30 nm). The position- and momentum-resolved net velocity $\bar{v}(x,k) = v(k)(f(k) - f(-k))$ shown in Supplementary Fig. 1, reveals the asymmetry of the Boltzmann distribution for $V_{gs} = 0.5$ V (Fig. 3b) occurs predominantly at the location and momentum at which the distribution function makes a transition from occupied (1) to unoccupied (0).

**Transfer characteristics**. We repeat the self-consistent calculation of the distribution function and the Poisson equation for different gate bias $V_{gs} = -0.5 \ldots 0.5$ V while fixing the drain-source bias to $V_{ds} = 0.1$ V and compute the current. This yields the transfer characteristic of the TI FET for different strengths of the scattering with imperfections, as shown in Fig. 4. With a gate-bias $V_{gs} \approx 0$ V, the current proceeds almost ballistically from source to drain since the edge states have limited back-scattering for all levels of imperfection scattering. With the application of a large positive or negative gate bias, carriers under the gate occupy states with an energy in the bulk conduction or valence band where back-scattering is severe because of the much larger overlap between wavefunctions. In the case of strong scattering the current decreases dramatically and an $I_{on}/I_{off}$ ratio of more than 2 orders of magnitude can be obtained.

Compared to conventional MOS FET transfer characteristics, the obtained transfer characteristics are similar: a gate bias in the range $V_{gs} = -0.5 \ldots 0$ V yields $n$MOS-like characteristics, whereas a gate bias in the range $V_{gs} = 0 \ldots 0.5$ V yields $p$MOS-like characteristics. The absolute gate bias ranges ($V_{gs} = -0.5 \ldots 0$ V and $V_{gs} = 0 \ldots 0.5$ V respectively) at which the $n$MOS or $p$MOS behaviour is exhibited, depend on the workfunction of the gate metal. The workfunction assumed in our simulation positions the gate Fermi level in the middle of the TI bandgap. However, as illustrated in Fig. 4b, appropriately choosing two alternernative gate metals with different workfunctions, an $n$TI FET and a $p$TI FET with their minimal current (off-state) at $V_{gs} = 0$ V can be obtained and complementary MOS (CMOS) logic circuits with low stand-by power can be designed.

Several important differences between TI FETs and conventional FETs exist in terms of behaviour with respect to imperfections, impact of tunnelling and threshold voltage variations. First, in Fig. 4, we observe that for the cases where there are little imperfections, the off-current dramatically increases while the impact on the on-current is much smaller. In the absence of imperfections, the transistor action is almost lost in our simulations as all electrons simply travel through the device ballistically. The TI FET behaviour with respect to imperfections is opposite to that in conventional FETs. Indeed, in conventional FETs off-current is minimally affected by scattering with imperfections whereas the on-current is severely limited by high levels of imperfection. Second, in conventional FETs tunnelling adversely affects the off-current. In TI FETs tunnelling does not adversely affect the off-current since scattering and not barriers are responsible for the reduced current in the off-state. Third, in CMOS based on conventional MOSFETs, operating at small voltages is problematic because of device-to-device threshold voltage variations. These threshold variations originate from the so-called $V_t$ roll-off associated with device length variations. In TI FETs, intrinsic process-independent scattering processes—and not channel length or doping—determine the threshold voltage. TI FETs will thus have improved immunity from $V_t$ roll-off and improved noise-margin tolerances than in the 'conventional' CMOS technology and can be operated at smaller voltages.

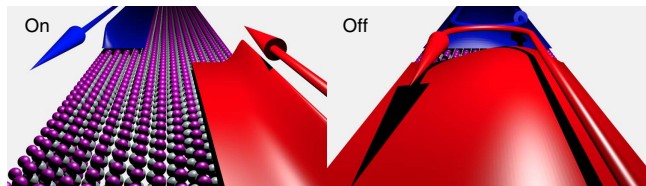

**Figure 2 | Schematic of a TI FET in the on-state and off-state.** In the on-state, current is carried by edge states and back-scattering is almost negligible in wide ribbons. In the off-state the states are no longer localized on the edge and scattering between states is dramatically increased. Only the spin-up component is illustrated. For spin-down, forward and backward transport will take place on the opposite edge.

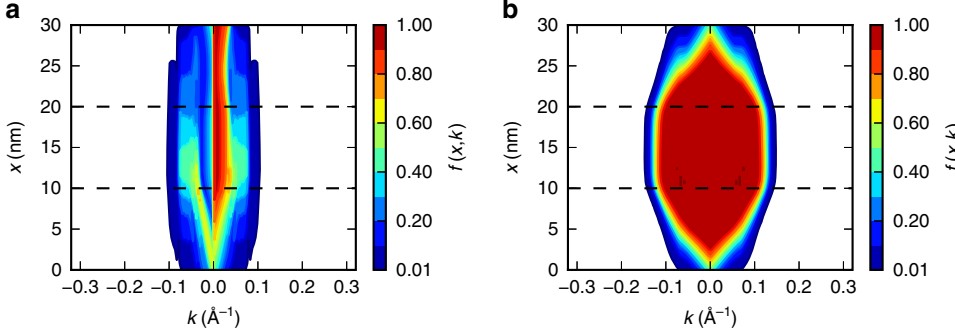

**Figure 3 | Boltzmann distributions.** Boltzmann distribution for the first conduction band in a TI FET with $V_{gs} = 0.1\,V$ (**a**) and $V_{gs} = 0.5\,V$ (**b**), for $V_{ds} = 0.1\,V$. The gate bias makes the charge density larger in the gate region (10–30 nm) compared with the source and drain regions. The strong asymmetry with respect to momentum of the distribution in **a** indicates a much larger current flow compared with the distribution in **b**, which is almost symmetric.

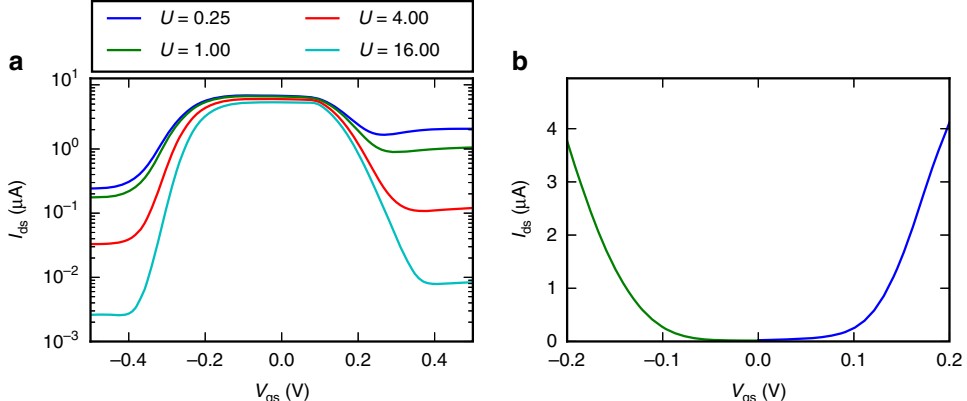

**Figure 4 | TI FET transfer characteristics.** (**a**) Transfer characteristics ($I_{ds} - V_{gs}$) of a TI FET obtained by solving the Boltzmann equation for $V_{ds} = 0.1\,V$ for different strengths of the scattering with imperfections $U = 0\ldots16\,eV\,nm$. Scattering is strong for $V_{gs} \approx -0.4\,V$ and $V_{gs} \approx 0.4\,V$ (off-state) and weak for $V_{gs} \approx 0\,V$ (on-state). For a TI with many imperfections, scattering reduces the off-current by more than two orders of magnitude while the on-current remains high. (**b**) Current for $V_{ds} = 0.2\,V$ with $U = 16\,eV\,nm$ on a linear scale with adjusted workfunctions. The $n$TI FET workfunction is decreased by 0.3 V while the $p$TI FET workfunction is increased by 0.43 V compared with the workfunction of the 2D TI. The current at $V_{gs} = 0\,V$ is $I_{off,n} = 23\,nA$ for the $n$TI FET and $I_{off,p} = 16\,nA$ for the $p$TI FET.

Apart from the imperfections and the phonons, alternative mechanisms and stronger scattering processes are likely to be active in these devices. The simulations performed with a larger $U$ (which we took to represent imperfections only in the preceding paragraphs) may also mimic these stronger or other scattering mechanisms. First, additional scattering processes, such as electron–electron scattering will increase the scattering rate. Second, even scattering with phonons may be significantly stronger than modelled in our simulations. Indeed, in lower dimensions (2D or 1D), phonons exhibit a parabolic rather than a linear dispersion (symmetry breaking yields massive Goldstone Bosons)[43–45]. In our case of ribbons, both the flexural (ZA) and the transverse (TA) phonon exhibit a parabolic dispersion. In our simulations we have not accounted for this potentially strong scattering process for the following reasons: the deformation potential for the flexural phonons (ZA) we obtained from first principles is very small for the particular TI under study (functionalized monolayer tin), and the transverse phonons (TA) are modelled using their linear bulk dispersion. So, in practice, even without scattering with imperfections, a large $I_{on}/I_{off}$ ratio can be obtained.

**Output characteristics.** In Fig. 5a we show the drain current for a gate bias $V_{gs} = -0.1\,V$, while varying the drain-source bias in the range $V_{ds} = 0\ldots0.5\,V$. At small drain-source bias ($V_{ds} < 0.1\,V$), the observed output characteristics are similar to those of the

MOS FET with an initial linear region governed by the quasi-ballistic transport through the edge states. On the other hand, for high drain bias, the output characteristics reveal a negative differential resistance. This can be explained by the observation that at large drain bias, the electrons can not travel through the entire device ballistically and scattering becomes inevitable. The region where the current can flow ballistically is limited by the TI bandgap. Indeed, we verify this by simulating a larger bandgap 2D TI for $V_{gs} = 0\,V$ and correspondingly see the maximum current at $V_{ds} = 0.26\,eV$ for the larger gap 2D TI compared with the maximum current at $V_{ds} = 0.13\,V$ for the smaller gap 2D TI in Fig. 5.

Because of the negative differential resistance (NDR), the TI FET does not provide enough drive current for voltages that significantly exceed the peak voltage. Operating in a conventional CMOS-like way would thus be limited to $V_{dd} \approx 0.2\,V$ for the smaller bandgap and $V_{dd} \approx 0.4\,V$ for the larger bandgap 2D TI. An alternative approach to enable operation at voltages beyond the peak voltage for small bandgap TIs, would be to exploit the NDR in a NDR-based logic configuration[8].

To get an estimate of the capacitance, we compute the total charge in the device $Q = \int dx \rho(x)$ at $V_{gs} = -0.35\,V$ and $V_{gs} = -0.15\,V$ with $V_{ds} = 0.2\,V$ for the small bandgap TI. The ratio between the charge variation $\Delta Q$ and the change of gate bias $\Delta V$ yields a capacitance of about 10.5 aF, which is small compared with conventional FET devices. The small capacitance is related

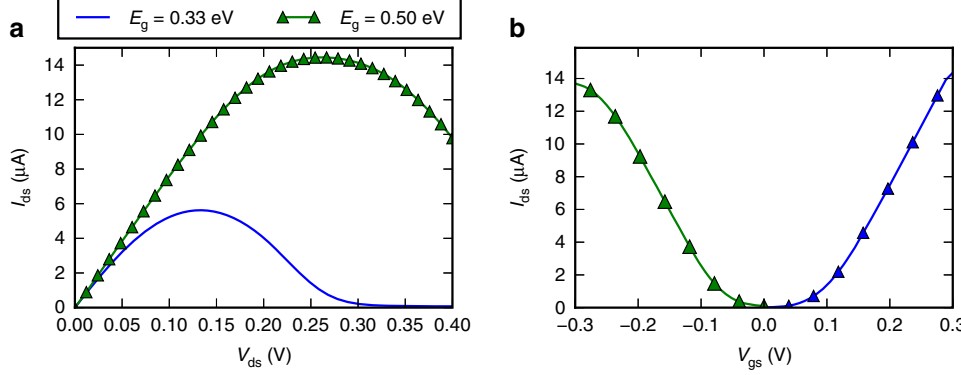

**Figure 5 | TI FET output characteristics. (a)** Output characteristics ($I_{ds} - V_{ds}$) of a TI FET for two different TIs, the first with $E_{g0} = 0.5$ eV resulting in a bandgap of 0.33 eV and the second with $E_{g0} = 1.0$ eV resulting in a bandgap of 0.5 eV. Accounting for the difference in the position of the valence maximum between both TIs, a gate bias of $V_{gs} = -0.1$ V is applied to the first and $V_{gs} = 0$.V to the second. The imperfection scattering parameter is set to $U = 16$ eV nm. At large drain bias in the on-state, negative differential resistance appears since scattering becomes inevitable. The peak at which the negative differential resistance occurs is proportional to the bandgap of the TI. **(b)** Similar to Fig. 4b for the larger bandgap 2D TI: $I_{ds}$ for $V_{ds} = 0.3$ V with $U = 16$ eV nm on a linear scale with adjusted workfunctions. The nTI FET workfunction is decreased by 0.3 V and has $I_{off,n} = 16$ nA while the pTI FET workfunction is increased by 0.6 V and has $I_{off,p} = 94$ nA.

to the low density of states of the edge states. The linear electron density in the edge states 1.93 eV$^{-1}$ nm$^{-1}$ combined with a short gate length make it so that only a few 10 s of electrons need to be displaced to switch the device.

**Benchmarking.** Based on the methodology presented in refs 46,47, where a 15 nm DRAM half-pitch was chosen, we make a crude estimate of the figures of merit for logic applications. We assume $V_{dd} = 0.2$ V and $I = 4$ μA, for the smaller bandgap TI FET, and $V_{dd} = 0.3$ V and $I = 14$ μA, for the larger bandgap TI FET. The capacitance is assumed $C = 10.5$ aF for both. We obtain an intrinsic switching speed for the smaller gap 2D TI (and larger gap 2D TI in parenthesis) $t_{int} = CV_{dd}/I \approx 0.52$ ps (0.22 ps) and an intrinsic energy per switching $E_{int} = CV_{dd}^2 \approx 0.42$ aJ (0.95 aJ). Taking the interconnect capacitance $C_{ic} = 37.8$ aF from ref. 46, interconnect delay is $t_{ic} = 0.7 C_{ic} \times V_{dd}/I \approx 1.32$ ps (0.61 ps) with an interconnect switching energy $E_{ic} = C_{ic} \times V_{dd}^2 \approx 1.5$ aJ (3.4 aJ). These figures show that the TI FET with the larger bandgap (TI FET 0.22 ps/0.95 aJ) is competitive with high-performance (HP) CMOS in terms of intrinsic delay (HP CMOS: 0.25 ps/19.63 aJ) and the homojunction tunnel FET (HomjTFET) in terms of power consumption (HomjTFET: 3.27 ps/0.98 aJ). The use of even larger bandgap 2D TIs than the one we simulated, such as the recently reported bismuthene with a 0.8 eV bandgap[33], would further improve the on-current and the intrinsic switching speed. In Fig. 6, we compare the results for the TI FET in a 32 bit arithmetic logic unit (ALU) based on the methodology presented in ref. 47 with those of other devices such as CMOS HP, CMOS LV, the BisFET, the interlayer tunnel FET (ITFET), and the metal-insulator transistion FET (MITFET). The results for the 32 bit ALU reveal a more significant trade-off between energy and speed when going to the larger supply voltage but confirm that the TI FET is competitive with other high-performance and low-power exploratory devices.

In the methodology from refs 46,47, off-current is not considered. Given the significant off-current that can be observed in Fig. 4b ($I_{off} \approx 0.02$ μA(0.09 μA)), the TI FET would have a large static power consumption $P_{static} = V_{dd}I_{off} = 4$ nW (27 nW) in a conventional CMOS setting. Active power consumption at a switching speed of $1/t_{int} = 0.6$ THz (1.1 THz) and assuming an activity factor of 1 would lead to $P_{active} = E_{ic}/t_{ic} = 0.8$ μW (3.9 μW) where active power dominates over static power by a factor by 200 (144). In a practical setting, however, circuit switching speed can not be set at $1/t_{int}$. Switching

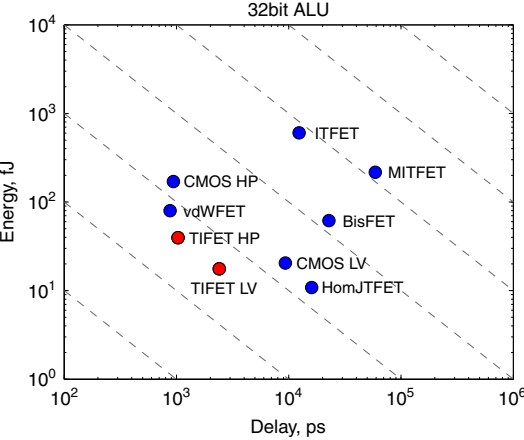

**Figure 6 | Benchmarking the TI FET versus other devices.** Switching delay versus energy for a 32-bit ALU determined using the methodology presented in ref. 47. The results for the smaller gap 2D TI FET with the 0.2 V supply voltage are indicated as TIFET LV and those for the 0.3 V supply voltage are indicated as TIFET HP.

more slowly than 1 GHz or at lower activity factors, static power consumption would inevitably come to dominate, compromising energy efficiency. Static power consumption can be reduced to some extent by changing the workfunction to reduce the off-current significantly and the on-current less significantly. Static power will also be reduced by increased scattering due to more imperfections or other scattering processes. Nevertheless, the off-current will always be large compared with low-standby power CMOS and the main target TI FET logic applications would thus be found in the realm of high-performance operation where high activity factors can be maintained.

## Discussion

We have modelled FETs using TI ribbons as active channel material by solving the Boltzmann equation accounting for ballistic transport and scattering while respecting Pauli's exclusion principle. The transfer characteristics ($I_{ds} - V_{gs}$) show that similar to CMOS, complementary TI FET logic is possible with the same kind of TI ribbon if two different gate metals are used. The off-current was shown to be more than two orders of magnitude below the on-current. We have argued that a

satisfactory on/off ratio can be maintained or improved in the presence of high concentrations of material defects or pronounced edge roughness. Comparing the TI FET performance to other devices, we have shown that the TI FET is competitive with high-performance CMOS in terms of speed and also competitive with TFETs in terms of energy consumption. The key to the exceptional performance of the TI FET is the small amount of charge in the channel and the ballistic current, both related to the topologically protected edge states. Our results motivate further research towards 2D TIs with large bandgaps to enable fabrication of room-temperature TI FETs for high-performance low-power nanoelectronics.

## Methods

**Band structure.** The band structure of the 2D TI under study is modelled after the theoretical band structure of iodine functionalized monolayer tin[28,38,39] (iodostannanane). We first compute the band structure from first principles using the Vienna ab initio Simulation Package (VASP)[40] using the Perdew–Burke–Ernzerhof functional[48]. Without accounting for spin–orbit coupling, the iodostannanane band structure is gapless with the valence and the conduction band touching each other at $\Gamma$ (Fig. 7a). Accounting for spin–orbit coupling opens up a bandgap, the conduction band minimum lies at $\Gamma$ while the top of the valence band has the shape of an inverted mexican hat around $\Gamma$ (Fig. 7b).

To facilitate the calculation of matrix elements for scattering and the band structure of ribbons with different widths, we use the $k \cdot p$-like BHZ Hamiltonian

$$H^{1/2}(\mathbf{K}) = \begin{bmatrix} \frac{E_{g0}}{2} - \frac{\hbar^2 K^2}{2m} + \frac{\hbar^2 K^2}{2m'} & \frac{\hbar p}{m_0}(k_y - ik_x) \\ \frac{\hbar p}{m_0}(k_y + ik_x) & -\frac{E_{g0}}{2} + \frac{\hbar^2 K^2}{2m} + \frac{\hbar^2 K^2}{2m'} \end{bmatrix},$$ (1)

$$H^{\mathrm{BHZ}} = H^{1/2}(k_x, k_y) \oplus H^{1/2}(k_x, -k_y)$$ (2)

rather than using the band structure obtained from first principles. We have written the BHZ Hamiltonian in the spirit of the two-band $k \cdot p$ Hamiltonian with a fundamental bandgap $E_{g0}$, an effective mass determining the curvature of both the valence and the conduction band $m$, and an effective mass determining the difference in curvature between the valence and the conduction band $m'$. The momentum matrix element $p$ measures the interaction between conduction and valence band and can be equivalently written in the form of an energy $E_p = 2p^2/m_0$. For the BHZ to be a TI, the sign of $E_{g0}m$ has to be positive. The parameters we use are $E_{g0} = 0.5$ eV, $E_p = 1.8$ eV, $m = 0.08m_0$ and $m' = 0.12m_0$ yielding an indirect bandgap of $E_g = 0.33$ eV.

It is well-known that first principles simulations using the Perdew–Burke–Ernzerhof functional underestimates the bandgap, also in the case of TIs (refs 29,30). As a simplified way to account for this, we also include simulations with an increased fundamental bandgap $E_{g0} = 1.0$ eV yielding an indirect bandgap of $E_g = 0.5$ eV.

The band structure of a TI ribbon is calculated by substituting $k_y \rightarrow \mathrm{id}/\mathrm{d}y$, setting a ribbon width $w$ and introducing a uniform mesh of $n_y$ points along $y$. Solving the Schrödinger equation yields the energies $E_j(k)$ and the ribbon wavefunctions $\phi_{kj}^\alpha(y)$, where $k_x$ is now simply denoted as $k$ and $\alpha$ is an index running over the four degrees of freedom of the bulk Hamiltonian given in equation (2).

**Boltzmann equation.** The essential physics of the TI FET consist of both ballistic transport in the on-state and strong scattering with phonons and imperfections in the off-state. We choose to model the TI FET using the Boltzmann equation over alternative approaches to study electron transport. Ballistic quantum transport approaches[49,50] are inappropriate since they do not account for scattering. The drift-diffusion-like approach we used previously[38] is incompatible with the ballistic limit. A non-equilibrium Green's function approach accounting for scattering is

computationally very expensive and state-of-the-art approaches are limited to a localized basis set for the band structure and the scattering interaction[51]. Within the non-equilibrium Green's function approach, respecting Pauli's exclusion principle in the presence of inelastic scattering is a daunting task[52] (In private communcation with the authors of ref. 51 confirmed that Pauli's exclusion principle can be violated using their approach.). The Pauli master equation[53] is only applicable in the weak-scattering regime.

The Boltzmann equation we solve is

$$\frac{\mathrm{d}E_j(k)}{\mathrm{d}\hbar k} \frac{\partial f_j(x,k)}{\partial x} + \frac{\mathrm{d}V(x)}{\hbar \mathrm{d}x} \frac{\partial f_j(x,k)}{\partial k}$$
$$+ \sum_{j'} \int \mathrm{d}k' \{ [1 - f_j(x,k)] S_{jj'}(k,k') f_{j'}(x,k')$$
$$- [1 - f_{j'}(x,k')] S_{j'j}(k',k) f_j(x,k) \} = 0$$ (3)

where $f_j(x,k)$ is the Boltzmann distribution function, $x$ the real-space and $k$ the reciprocal coordinate in a range $[0,L]$ and $[-k_{max}, k_{max}]$, respectively. The simulated region has a length $L$ and $k_{max}$ is the largest $k$-value under consideration. The index $j$ denotes the subband of the distribution function. The rate for an electron to make a transition from an initial state $j'k'$ to a final state $jk$ is measured by $S_{jj'}(k,k')$.

The potential $V(x)$ in equation (3) is obtained from the solution of the Poisson equation

$$\nabla^2 V_{\mathrm{p}}(\mathbf{r}) = \frac{\rho_{\mathrm{p}}(\mathbf{r})}{\epsilon}.$$ (4)

The subindex p is introduced to distinguish between the Poisson and the Boltzmann equation grids. Rather than accounting for the atomistic dielectric response[54], for simplicity we have chosen a uniform dielectric constant $\epsilon$ in our simulations. Taking the simplified uniform dielectric approach presents a minor approximation since the charge density (and the right hand side of equation (4)) is generally small in our simulations.

Whereas the Boltzmann equation is solved only along the transport direction, the Poisson equation is solved in two dimensions $(x, z)$ where $x$ is the transport direction, the same direction the Boltzmann equation is solved in, and $z$ is the direction perpendicular to the channel. The nanoribbon channel is taken to have a thickness $t$ and extends from $z \in ]-t/2, t/2[$. A double-gate configuration is simulated with a gate dielectric in the regions $|z| \in [t/2, t/2 + t_{ox}[$, and the boundary condition $V_{\mathrm{p}}(x, z) = V_g$ for the two gate electrodes is applied at $|z| = t/2 + t_{ox}$ and $|x - L/2| < L_{\mathrm{gate}}/2$.

The Boltzmann and the Poisson equations are coupled and the potential of the former is taken to be related to the potential of the latter through $V(x) = V_{\mathrm{p}}(x, 0)$. The charge is obtained from the Boltzmann distribution through

$$\rho(x) = e \sum_{j \in v} \int \frac{\mathrm{d}k}{2\pi} [1 - f_j(x,k)] - e \sum_{j \in c} \int \frac{\mathrm{d}k}{2\pi} f_j(x,k)$$ (5)

where $e$ is the elementary charge and $v$ and $c$ are the set of all the valence and conduction band indices, respectively. In TI ribbons, the valence (conduction) bands are all bulk-like valence (conduction) bands together with the band formed by the TI edge states with the lower (higher) energy. The charge in equation (5) is converted into the 3D charge density required by the Poisson equation by setting $\rho_{\mathrm{p}}(x, z) = \rho(x)/(wt)$ for $|z| < t/2$ and we assume there is no fixed charge in the gate dielectric so $\rho_{\mathrm{p}}(x,z) = 0$ otherwise.

We discretize the Boltzmann distribution function $f_i(x, k)$ on a uniform $n_x \times n_k$ real-space and reciprocal-space grid. The reciprocal-space differential operator $\mathrm{d}/\mathrm{d}k$ is discretized using a central difference scheme $\mathrm{d}f(x,k)/\mathrm{d}k = [f(x, k + \Delta k) - f(x, k - \Delta k)]/(2\Delta k)$ and periodic boundary conditions are applied at $-k_{max}$ and $k_{max}$. The real-space differential operator $\mathrm{d}/\mathrm{d}x$ is discretized using a finite element scheme with $n_x + 1$ nodes and $n_x$ elements so that $\mathrm{d}f(x, k)/\mathrm{d}x|_{x = x + \Delta x/2} = [f(x + \Delta x, k) - f(x, k)]/\Delta x$ and $f(x, k)|_{x = x + \Delta x/2} = [f(x + \Delta x, k) + f(x, k)]/2$. At the outer nodes, boundary conditions are introduced such that the injected Boltzmann distribution function is in thermal equilibrium.

Because of the Pauli exclusion principle, equation (3) is a non-linear equation and we apply Newton's method to solve it iteratively. Denoting the left hand side of equation (3) with $F(f)$, the Boltzmann equation is solved when $||F(f)|| = 0$. Fortunately, equation (3) admits an exact calculation of the Jacobian $\mathcal{J} = \mathrm{d}F(f)/\mathrm{d}f$. Representing the differential operators with matrices, $\mathcal{J}$ is a sparse matrix and the update to the distribution function $\Delta f = \mathcal{J}^{-1} F$ can be computed efficiently through sparse LU factorization[55].

The current can be computed as

$$J(x) = e \sum_j \int \frac{\mathrm{d}k}{2\pi} \frac{\mathrm{d}E_j(k)}{\hbar \mathrm{d}k} f_j(x,k)$$ (6)

and on convergence, the current is continuous throughout the device. To further analyse the current distribution, we can define a position-, band- and momentum-dependent net velocity

$$\bar{v}_j(x, k) = \sum_j \frac{\mathrm{d}E_j(k)}{\hbar \mathrm{d}k} [f_j(x,k) - f_j(x, -k)]$$ (7)

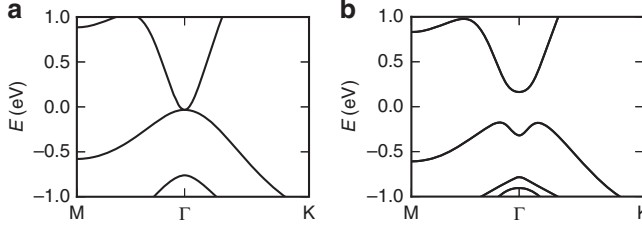

**Figure 7 | Bulk topological insulator band structure calculated from first principles.** . Band structure without spin–orbit coupling (**a**) and with spin–orbit coupling (**b**).

so that $J(x) = e \sum_j \int_0^{k_{max}} dk/(2\pi)\bar{v}_j(x,k)$. In Supplementary Fig. 1, we show the position- and momentum-dependent net velocity for the simulation of $V_{gs} = 0.5$ V and $V_{ds} = 0.1$ V for which the Boltzmann distribution is shown in Fig. 3b of the results section.

**Scattering.** To account for the electron–phonon interaction, we include scattering with phonons through a deformation potential approximation

$$S^{em,abs}_{jj'}(k,k') = \frac{DK^2}{2\rho_{1D}\omega}\left[\frac{1}{2}\mp\frac{1}{2}+N(\hbar\omega)\right]|M_{jj'}(k,k')|^2\delta(E_j(k)-E_{j'}(k')) \quad (8)$$

where $\omega$ is the phonon angular frequency, $\rho_{1D}$ is the charge per unit length, $N(\hbar\omega)$ the Bose–Einstein distribution function and

$$M_{jj'}(k,k') = \sum_\alpha \int dy\, \phi_k^{\alpha*}(y)\phi_{k'}^\alpha(y) \quad (9)$$

is the overlap between the wavefunctions. Equation (8) assumes the bulk phonons of the 2D TI can be used. However, as discussed in the 'Results' section, quantization of phonons will increase the scattering rates[56]. The deformation potentials $DK$ are calculated from first principles[40] as explained in refs 26,42. For elastic scattering with longitudinal and transverse acoustic phonons and intraband back-scattering (so that $k' = -k$, $DK = \Delta q$ and equation (8) simplifies to

$$S^{em,abs}_{jj'}(k,k') = \Delta^2\frac{kT}{\hbar 2\rho_{1D}v_s^2}\frac{|M(k,k')|^2}{|dE/dk|}\delta(k+k'). \quad (10)$$

To include scattering with imperfections, we assume the imperfection scattering Hamiltonian can be described by a delta-like potential perturbation

$$H^{imp} = A\delta(x-x_0)\delta(y-y_0)\delta_{\alpha\alpha_0} \quad (11)$$

where $A$ relates to the strength of the perturbation and will depend on the kind of imperfection. The magnitude of the matrix element with such a delta-like potential is

$$\left|\langle j'k'|H^{imp}|jk\rangle\right| = \left|A\phi_{jk}^{\alpha_0}(y_0)\phi_{j'k'}^{\alpha_0}(y_0)/l\right| \quad (12)$$

(to give equation (12) units of energy, we introduced a length $l\to\infty$ to normalize the plane-waves along the $x$-direction, that is, $\psi(x,y) = e^{ikx}/\sqrt{l}\phi(y)$). The scattering rate associated with one imperfection is

$$\frac{1}{\tau_{imp}} = \frac{2\pi A^2}{\hbar l^2}\left|\phi_{jk}^{\alpha_0}(y_0)\phi_{j'k'}^{\alpha_0}(y_0)\right|^2\delta(E_{jk}-E_{j'k'}). \quad (13)$$

Assuming a set of delta-like potentials is randomly placed throughout the TI ribbon with an areal density $N_{imp}$, the number of scatterers is $N_{imp}wl$. Now inspecting equation (3) and relying on the fact that in limit $l\to\infty$, $2\pi/l\sum_k\to\int dk$, the imperfection scattering rate for the Boltzmann equation (equation (3)) is

$$S^{imp}_{jj'}(k,k') = \frac{U^2\left|M^{imp}_{jj'}(k,k')\right|^2}{\hbar\left|dE_j/dk\right|}\delta(k+k')\delta_{jj'} \quad (14)$$

for intraband back-scattering ($k\to -k$, as reflected by $\delta(k+k')$) where

$$\left|M^{imp}_{jj'}(k,k')\right|^2 = \sum_\alpha\int_0^w dy\left|\phi_{jk}^\alpha(y)\phi_{j'k'}^\alpha(y)\right|^2 \quad (15)$$

and $U = \sqrt{N_{imp}}A$. The strength of the scattering with imperfection thus depends on their density and the kind of imperfection.

To speculate on the possible magnitude of the imperfection strength, we calculating the matrix element for the case of a dangling bond using first principles and obtain a value of $A = |\langle j'k'|U_0-U_{dangling}|jk\rangle|WL = 14$ eV nm[2]. As an upper limit, at large concentrations of $N_{imp}\approx 0.5$ nm$^{-2}$ (one imperfection in one in every 10 unit cells), $U$ would then reach values on the order of 10 eV nm. This motivates our choice of $U = 0.25$ eV nm $\to 16$ eV nm in the main text. Also, as mentioned in the main text, other scattering processes such as edge roughness and electron–electron scattering could also lead to large scattering rates in these materials and could also be mimicked by a certain value of $U$.

Unlike most scattering processes, equation (14) has no $q$-dependence apart from the overlap of the wavefunctions. For example, LER has a $q$-dependence of the form $M_{LER}\propto U/(1+0.5q^2\Lambda^2)^{3/2}$ (for exponential correlation) with $U = DV\Delta$, where $DV$ is the depth/height of the scattering potential and $\Delta$ the step height of the roughness, and $\Lambda$ is the LER correlation length. However, LER has minimal $q$-independence when $q\Lambda$ is much $< 1$. With an electron energy $E = \hbar v_F$, assuming a typical value $\Lambda = 1$ nm and a Fermi velocity, $v_F = 5\times 10^5$ m s$^{-1}$, this implies that up to electron kinetic energies of about 0.2 eV ($k\Lambda = 1$ at 0.328 eV), our $q$-independent imperfection scattering matrix process would be identical to the LER-scattering process. Taking a $DV\approx 1Ry = 13.6$ eV and a anti-correlated roughness with step height $\Delta = 0.2$ eV would yield a $U = 5.4$ eV nm which is also on the order of 10 eV.

**Data availability.** The data that support the findings of this study are available from the corresponding author upon request.

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

## Acknowledgements

We acknowledge the support of Nanoelectronics Research Initiative's (NRI's) Southwest Academy of Nanoelectronics (SWAN). The authors thank Kristof Moors, Christopher Hinkle, Robert Wallace, Luigi Colombo and Dmitri Nikonov for fruitful discussions.

## Author contributions

W.G.V. developed the theory and code, performed the simulations and wrote the manuscript. M.V.F. was involved in the description of the electron–phonon and the imperfection/edge roughness scattering and improved the manuscript text.

## Additional information

**Competing financial interests:** The authors declare no competing financial interests.

