## [Peer Review File · Nature Communications]

Reviewer #1 (Remarks to the Author):

The authors have theoretically simulated a FET based on TI nano ribbons. They have employed suitable transport properties of these ribbons for designing a practical nano-FET. However, it seems that the manuscript could be much improved if they consider some issues:

1- Since in the introduction, the transport properties are reviewed, it will be much better if the transport properties of 2D-TI can be compared with 3D- case.

2- Since from the device engineering perspective, the gate control ability on the transistor current is very important, the authors should show that which TI creates better performance: 2D- nano ribbons or 3D-TI?

3- It is not obvious that which physical mechanism affects the edge defects and leads to the current drop in Fig.4.

4- It is claimed that in Fig.4 the gate voltage increases the edge defects and thus results in the current drop. However, I could not understand that why the conduction is higher under the gate in Fig.3. Please clarify.

5- Fig.3(a) should be more justified. It is claimed later in Fig.5 that the drain voltage increase results in more edge defects and thus more current drop. If this is so, why this impact can not be seen on the drain region in Fig.3(a)?

Reviewer #2 (Remarks to the Author):

The authors propose a field effect transistor based on 2D nanoribbons of topological insulators.

The big advantage of the proposed device is the possibility to operate and have interesting performance also in the presence of high concentration of defects in the 2D channel, such as vacancies and edge roughness.

In the ON state, current flows in edge states and therefore transport is ballistic even if a high concentration of defects is present. In the OFF state, current flows in states in the bulk valence or conduction band, and scattering suppresses current.

The device was already proposed in a 2014 IEDM paper by the same authors, but investigated only in terms of drift-diffusion transport, i.e., assuming quasi-equilibrium conditions. In this manuscript, device operation and performance are evaluated by means of self-consistent 1D simulations of BTE and 2D simulations of Poisson equations.

I appreciate the interesting physics but at the same time completely disagree on the assessment of device performance and on the evaluation of the implications for device operation.

For the device to be used in CMOS-like digital logic, the gate modulation $\Delta V_{GS} = V_{GSon} - V_{GSoff}$ must be equal to V_{DS} . However, it is pretty clear from Fig. 5 that for $\Delta V_{GS} = V_{DS} = 0.3$ V there is negligible current modulation, since in non equilibrium conditions, electrons at some point in the channel have an energy in the bulk conduction band and therefore are subject to strong scattering. The authors clearly highlight this aspect. Therefore the logic conclusion would then be: the device does not work.

Instead, the authors present an optimistic conclusion, because they use the metrics t_{int} and E_{int} (lines 164 -171) using a ΔQ computed with $\Delta V_{GS} = 0.3$ V and $V_{DS} = 0.1$ V. In this situation the current modulation is very good, but the metrics t_{int} and E_{int} have physical meaning only for $V_{DS} = 0.3$ V. Their promising figures of merit are due to the wrong use of the metrics. A proper use of the metrics (computed with $V_{DS} = 0.3$ V) would provide very poor figures of merit, at least for this particular device.

Clearly, the authors have not demonstrated "high performance" or "exceptional performance" in

2D TI FETs as they claim.

For this critical aspect, I do not recommend this paper for publication on Nature Communications. The same manuscript, with a realistic assessment of device operation, correcting the overoptimistic simulation of the IEDM 2014 paper, could be suitable for a more specialised journal.

Minor comment:

In Fig. 3: V_{ds} is indicated to be 0.1 V in the main text and 0.3 V in the figure caption.

Reviewer #3 (Remarks to the Author):

The manuscript "Imperfect Two-dimensional Topological Insulator Field-effect Transistors" reports on the possibility of using two-dimensional topological insulators as channel materials for field-effect transistors (FETs).

In particular the authors put forward a possible mechanism to switch the device that actually exploits the inevitable sources of scattering in the system (edges, defects, phonons, etc.) and seems more feasible with respect to previous proposals based on unrealistically large electric fields.

The underlying idea is that the off state can be induced by using the gate voltage to drive the Fermi level outside the bulk gap, where states are no longer protected from backscattering and the current drops by several orders of magnitude.

The proposal is interesting and the results are scientifically sound.

My main concern is that the authors have already put forward this idea in a previous paper (Ref. [37]), greatly reducing the novelty and originality of the present manuscript.

Here the main difference is the theoretical approach to compute the transport properties of these FETs (Boltzmann transport equation instead of drift-diffusion model) while the main conclusions seem to be the same.

For this reason I would consider this manuscript more appropriate for a more specialized journal unless the authors will be able to argue convincingly the opposite.

Another important methodological concern is the following:

In the short channel regime tunneling should be extremely relevant but it is not accounted for in the semiclassical Boltzmann approach adopted here.

This would increase the off current and thus affect detrimentally the (already not satisfactory) on/off ratio, casting some doubts on the effectiveness of the switching mechanism proposed.

Can the authors comment on this?

Reviewer #1

Question 1: Since in the introduction, the transport properties are reviewed, it will be much better if the transport properties of 2D-TI can be compared with 3D- case.

Answer 1: *This is an important point raised by the reviewer since 2D TIs have many important advantages over 3D TIs for logic applications. We have added to the introduction “Three-dimensional TIs also have surface states whose presence is protected against imperfections. However, for transistor applications 3D TIs have severe disadvantages: i) A 3D TI will inevitably suffer from shunting paths through the bulk and through surfaces other than the surface on which the device is fabricated, ii) the surface states of 3D TIs are effectively metallic making it hard to significantly move the Fermi level, and iii) while the 3D TI surface states are also spin-polarized, making them possible candidates for spin-based memory devices, conduction is not ballistic in 3D surface states.”*

Question 2: Since from the device engineering perspective, the gate control ability on the transistor current is very important, the authors should show that which TI creates better performance: 2D- nano ribbons or 3D-TI?

Answer 2: *Our answer to Question 1 should address this concern.*

Question 3: It is not obvious that which physical mechanism affects the edge defects and is leads to the current drop in Fig.4.

Answer 3: *The physical mechanism that leads to the current drop in Fig. 4 is the increased scattering with imperfections. A practical example of such a process is line-edge roughness which we have previously shown in Ref. 14 to be very detrimental for “conventional” FET operation in graphene nanoribbons. We believe some confusion may come from Fig. 3 as indicated in our next Answer*

Question 4: It is claimed that in Fig.4 the gate voltage increases the edge defects and thus results in the current drop. However, I could not understand that why the conduction is higher under the gate in Fig.3. Please clarify.

Answer 4: *The current is not higher under the gate, current is continuous through the device. The apparent increase in Fig. 3b of the previous submission (Fig. 2 supplemental current submission) is because the net velocity has a higher peak value under the gate. To maintain clarity of the contour plot, values less than 100 m/s are not shown although they are included in the calculation of the current. The total current at each position is thus continuous. The point of Fig. 3b of the previous submission (Fig. 2 supplemental*

current submission) is to show that the asymmetry in the distribution function depends on the gate bias and in turn determines the scattering. We agree with the reviewer that Fig. 3b of the previous submission was confusion and have moved it to the supplemental information. Instead, we have introduced the Boltzmann distribution for $V_{gs} = 0.1$ V, where the the asymmetry in the distribution is apparent. We have also added “Current (integral of the net velocity over k) is continuous as a function of x .” in the caption.

Question 5: Fig.3(a) should be more justified. It is claimed later in Fig.5 that the drain voltage increase results in more edge defects and thus more current drop. If this is so, why this impact can not be seen on the drain region in Fig.3(a)?

Answer 5: Again, we apologize for the lack of clarity in our presentation and hope our previous two answers clarify the situation. Fig. 3 (a) shows that the asymmetry is large for $V_{gs} = 0.1$ V indicating a large current can flow because there is limited scattering with imperfections (edge defects). Fig. 3 (b) shows that the asymmetry is small for $V_{gs} = 0.5$ V indicating a small current because there is a large amount of scattering with imperfections (edge defects).

Reviewer #2 (Remarks to the Author):

The authors propose a field effect transistor based on 2D nanoribbons of topological insulators.

The big advantage of the proposed device is the possibility to operate and have interesting performance also in the presence of high concentration of defects in the 2D channel, such as vacancies and edge roughness.

In the ON state, current flows in edge states and therefore transport is ballistic even if a high concentration of defects is present. In the OFF state, current flows in states in the bulk valence or conduction band, and scattering suppresses current.

The device was already proposed in a 2014 IEDM paper by the same authors, but investigated only in terms of drift-diffusion transport, i.e., assuming quasi-equilibrium conditions. In this manuscript, device operation and performance are evaluated by means of self-consistent 1D simulations of BTE and 2D simulations of Poisson equations.

I appreciate the interesting physics but at the same time completely disagree on the assessment of device performance and on the evaluation of the implications for device operation.

Question 5: For the device to be used in CMOS-like digital logic, the gate modulation $\Delta V_{GS} = V_{GSon} - V_{GSoff}$ must be equal to V_{DS} . However, it is pretty clear from Fig. 5 that for $\Delta V_{GS} = V_{ds} = 0.3$ V there is negligible current modulation, since in non equilibrium conditions, electrons at some point in the channel have an energy in the bulk conduction band and therefore are subject to strong scattering. The authors clearly highlight this aspect. Therefore the logic conclusion would then be: the device does not work.

Answer 5: *We thank the reviewer for his concerns about the device operation: a critical attitude towards new device concepts is certainly required in the community. If the logical conclusion were “that the device does not work”, the paper would indeed not warrant publication. However, we respectfully disagree with the reviewers’ assessment. We should take the blame for our lack of clarity that we believe is at the origin of the reviewer’s comment.*

We have chosen the value $V_{DS}=0.3$ V to obtain some performance metric for the device while the device we had simulated did not provide sufficient drive current at this voltage. However, our results make it clear that the device can operate in conventional CMOS way with $V_{DS} = 0.2$ V, with good performance as well. It would be sufficient to select two gate metals with suitable work-functions to realize this mode of operation. Another notable feature exhibited by our device is the immunity from “ V_T roll-off” that would allow us to implement this CMOS operation with better immunity from noise-margin tolerances than in the ‘conventional’ CMOS technology. This is due to the fact that the intrinsic, process-independent scattering processes -- and not channel length or doping -- determine the threshold voltage in TI FETs.

We should also add that the negative differential resistance (NDR) seen at $V_{GS} > 0.2$ V is caused by the small gap we have calculated for monolayer tin. We have chosen our topological insulator parameters based on the results of DFT-GGA results for functionalized monolayer tin, but DFT-GGA is known to underestimate the bandgap (also in TIs as shown in Ref. 30). In particular, the focus of our manuscript lies with the general concept of the TI FET and the ability to deal with imperfections. A recent paper shows the experimental demonstration of a 2D TI (monolayer Bi, ‘bismuthene’) with 0.8 eV bandgap (<https://arxiv.org/abs/1608.00812>). In a 0.8 eV bandgap 2D TI, there will be no NDR or reduction of the current at $V_{DS}=0.3$ V. In this sense, the figure of merit in the paper could be considered a conservative estimate. As a demonstration that the NDR vanishes for larger gap 2D TIs, we have included the simulation of a 0.5 eV bandgap material in the supplemental material.

Significant changes to the paper and the supplementary material have been made to address all of these concerns.

Question 6: Instead, the authors present an optimistic conclusion, because they use the metrics t_{int} and E_{int} (lines 164 -171) using a ΔQ computed with $\Delta V_{GS} = 0.3$ V and $V_{ds} = 0.1$ V. In this situation the current modulation is very good, but the

metrics t_{int} and E_{int} have physical meaning only for $V_{\text{ds}} = 0.3$ V. Their promising figures of merit are due to the wrong use of the metrics. A proper use of the metrics (computed with $V_{\text{ds}}=0.3$ V) would provide very poor figures of merit, at least for this particular device.

Clearly, the authors have not demonstrated “high performance” or “exceptional performance” in 2D TI FETs as they claim.

Answer 6: *We have calculated the metric for a supply voltage of 0.2 V for the small bandgap TI (corresponding to the GGA bandstructure of functionalized monolayer tin) and for a supply voltage of 0.3 V for a device with a larger peak current. We have also extended our investigation using the methodology put forward in Ref. 37, generally accepted by the community. In Fig. 6, we show the switching delay and energy for a 32-bit Arithmetic Logic Unit (ALU). As can be expected, a trade-off between power consumption and performance is observed based on the supply voltage is observed but still showing that TI FET is competitive with other high-performance and low-power exploratory devices*

Question 7: For this critical aspect, I do not recommend this paper for publication on Nature Communications. The same manuscript, with a realistic assessment of device operation, correcting the overoptimistic simulation of the IEDM 2014 paper, could be suitable for a more specialised journal.

Answer 7: *We agree that the 2014 IEDM manuscript did not make a realistic assessment of device operation while the current manuscript does. Also, given the 4-page limitation, the lack of peer review of the paper, and the preliminary nature of the results, the IEDM 2014 publication should be considered an abstract rather than a regular paper. Important, however, is the fact that our current manuscript shows that high-performance FETs made out of 2D topological insulators with many imperfections (such as line edge roughness and defects) can be made. This finding is important for the community which is struggling with material imperfections. No benchmarking was performed either. For this reason, it justifies publication in Nature Communications, and the consideration about the imperfections was not discussed nor realized in the 2014 IEDM paper.*

Question 8: Minor comment:

In Fig. 3: V_{ds} is indicated to be 0.1 V in the main text and 0.3 V in the figure caption.

Answer 8: *We thank the reviewer and have corrected the error; the number was incorrectly mentioned in the figure caption.*

Reviewer #3 (Remarks to the Author):

The manuscript "Imperfect Two-dimensional Topological Insulator Field-effect Transistors" reports on the possibility of using two-dimensional topological insulators as channel materials for field-effect transistors (FETs).

In particular the authors put forward a possible mechanism to switch the device that actually exploits the inevitable sources of scattering in the system (edges, defects, phonons, etc.) and seems more feasible with respect to previous proposals based on unrealistically large electric fields.

The underlying idea is that the off state can be induced by using the gate voltage to drive the Fermi level outside the bulk gap, where states are no longer protected from backscattering and the current drops by several orders of magnitude.

Question 9: The proposal is interesting and the results are scientifically sound.

My main concern is that the authors have already put forward this idea in a previous paper (Ref. [37]), greatly reducing the novelty and originality of the present manuscript.

Here the main difference is the theoretical approach to compute the transport properties of these FETs (Boltzmann transport equation instead of drift-diffusion model) while the main conclusions seem to be the same.

For this reason I would consider this manuscript more appropriate for a more specialized journal unless the authors will be able to argue convincingly the opposite.

Answer 9: *In Ref. [37] no discussion was made of the impact of imperfections (e.g., surface roughness). The current manuscript highlights the importance of scattering with imperfections and the ability of the TI FET to operate under these circumstances. We believe this is a new realization and deserves dissemination to a wide audience. Imperfections are an important point in our current paper and its conclusion. In Ref. [37], the claims of energy-efficiency and high performance were very speculative while the current manuscript presents clear evidence thereof with benchmarking results. We have added Fig. 6 where switching delay versus energy for a 32-bit Arithmetic Logic Unit (ALU) are shown. Additionally, Ref. [37] should be considered an abstract to an oral presentation, usually not considered duplicate publications, rather than a regular scientific paper whose results can be reproduced by the community.*

Question 10: Another important methodological concern is the following:

In the short channel regime tunneling should be extremely relevant but it is not accounted for in the semiclassical Boltzmann approach adopted here.

This would increase the off current and thus affect detrimentally the (already not satisfactory) on/off ratio, casting some doubts on the effectiveness of the switching mechanism proposed.

Can the authors comment on this?

Answer 10: *We thank the reviewer for this comment, it is an item we had not considered yet. Fortunately, since the TI FET does not rely on the band gap to stop current in the off-*

state, tunneling will not adversely affect the off-current. It should therefore be considered a strength and we have added to the manuscript “Second, in conventional FETs tunneling adversely affects the off-current. In TI FETs tunneling does not adversely affect the off-current since scattering and not barriers are responsible for the reduced current in the off-state.”

Reviewer #1 (Remarks to the Author)

Authors have revised the manuscript accordingly and I am now convinced that it can be published in "Nature Communications".

Reviewer #2 (Remarks to the Author)

The authors have submitted a largely revised manuscript in which they have addressed my criticisms to the previous version.

First, they have modified the supply voltage used for the assessment of device operation to 0.2 V (it was 0.3 V). Second, they have simulated a device with a larger gap in the channel ($E_g=0.33$ eV), that can operate properly with a supply voltage of 0.3 V.

The authors also write that the TIFET can work with a small supply voltage because it has a more robust V_T than other FETs, since it is dependent on "intrinsic, process independent scattering processes". However, a much more important disadvantage of the small-gap TIFET and of the 0.2 supply voltage is the high current in the off state I_{off} , which appears in Fig. 4b to be quite high. On the other hand, the larger gap TIFET should have a smaller current I_{off} (never discussed).

Then the authors add a comparison in terms of performance metrics with other exploratory devices, following the methodology of Ref. 47. I appreciate the intention, but I want to stress a drawback of the methodology, in that it does not include directly the static power consumption (this issue is also mentioned in Ref. 47). Indeed, one real improvement of the methodology would be - instead of choosing the best I_{on} for each device - to choose the best I_{on} at I_{off} parity (as for example it is done in the ITRS), so that at least the static power of compared technologies is similar. In the case of the TIFET for example, it would appear clearly that the small gap TIFET has a terrible static power (the large-gap would be better, but there are no data in the manuscript).

In conclusion, I think that the revised manuscript has been significantly improved, and I suggest publication on Nature Communications if the following improvements are made:

1. Comment on the type and quantity of defects required to have $U = 16\text{eVnm}$.
2. Mention the I_{off} of the small-gap and large-gap TIFETs obtained the chosen p- and n-FET gate workfunctions.
3. Adjust Fig. 4b so that for $V_{ds} = V_{gs} = 0.2$ V the current is indeed 4 microamps as considered in the text and in the performance assessment (from Fig. 4b it seems 6.3 microamps because probably a different assumption has been made on the gate workfunctions)
4. Add a Figure similar to Fig. 4b for the large-gap TIFET at $V_{ds} = 0.3$ V, so that one can appreciate the I_{off} .
5. Comment the issue of the static power consumption (with quantitative figures) when discussing performance metrics.

Reviewer #3 (Remarks to the Author)

I'm satisfied with how the authors have addressed the criticisms by me and the other reviewers, I thus support the publication of the present manuscript in Nature Communications.

RESPONSE TO THE REVIEWERS

We would like to express our appreciation to the reviewers for their careful reading of our manuscript.

Reviewer #1 wrote: *“Authors have revised the manuscript accordingly and I am now convinced that it can be published in "Nature Communications"”*.

Response: We thank the reviewer again for his/her comments.

Reviewer #2 wrote:

“The authors have submitted a largely revised manuscript in which they have addressed my criticisms to the previous version.

First, they have modified the supply voltage used for the assessment of device operation to 0.2 V (it was 0.3 V). Second, they have simulated a device with a larger gap in the channel ($E_g=0.33$ eV), that can operate properly with a supply voltage of 0.3 V.

The authors also write that the TIFET can work with a small supply voltage because it has a more robust V_T than other FETs, since it is dependent on “intrinsic, process independent scattering processes”. However, a much more important disadvantage of the small-gap TIFET and of the 0.2 supply voltage is the high current in the off state I_{off} , which appears in Fig. 4b to be quite high. On the other hand, the larger gap TIFET should have a smaller current I_{off} (never discussed).

Then the authors add a comparison in terms of performance metrics with other exploratory devices, following the methodology of Ref. 47. I appreciate the intention, but I want to stress a drawback of the methodology, in that it does not include directly the static power consumption (this issue is also mentioned in Ref. 47). Indeed, one real improvement of the methodology would be - instead of choosing the best I_{on} for each device - to choose the best I_{on} at I_{off} parity (as for example it is done in the ITRS), so that at least the static power of compared technologies is similar. In the case of the TIFET for example, it would appear clearly that the small gap TIFET has a terrible static power (the large-gap would be better, but there are no data in the manuscript).

In conclusion, I think that the revised manuscript has been significantly improved, and I suggest publication on Nature Communications if the following improvements are made:”

Response: We have made the suggested changes as itemized below and thank the reviewer for pointing out some shortcomings in our previous versions.

Question 1. *“Comment on the type and quantity of defects required to have $U = 16\text{eVnm}$.”*

Answer 1: An exact quantification of the type of defects and their scattering is beyond the scope of the manuscript. To our knowledge it is even an open question in the research community. Nevertheless, the question is a relevant question and in the last 3 paragraphs of the supplementary material, we have formalized our derivation of the imperfection scattering rate. We have also made a crude estimate of the concentration of imperfections. We show that $U=A N_{\text{imp}}^{1/2}$ where A is the strength of the scattering with a single imperfection and N_{imp} is the concentration of impurities. From first principles, on a 6nm ribbon we estimate the value of A as 14 eVnm^2 for a dangling bond (topological edge states are maintained in our first principles study showing that TI FET operation is still possible at this defect concentration). With $A=14\text{ eVnm}^2$ and a concentration of $0.5/\text{nm}^2$ (one defect in every 10 unit cells), this yields $U=10\text{ eVnm}$. For line-edge roughness we compute values on the order of $U=5\text{ eVnm}$. We believe this shows that our simulations ranging from $U=0.25\text{ eV nm}$ to 16 eVnm capture a realistic range. Both $U=4\text{ eVnm}$ and $U=16\text{ eVnm}$ show good TI FET behavior in Fig. 4a.

Question 2. *“Mention the I_{off} of the small-gap and large-gap TIFETs obtained the chosen p- and n-FET gate workfunctions.”*

Answer 2: We have added to the caption of Fig. 4 “The current at $V_{\text{gs}}=0\text{ V}$ is $I_{\text{off},n}=23\text{ nA}$ for the nTI FET and $I_{\text{off},p}=16\text{ nA}$ for the pTI FET.”

Question 3. *“Adjust Fig. 4b so that for $V_{\text{ds}} = V_{\text{gs}} = 0.2\text{ V}$ the current is indeed 4 microamps as considered in the text and in the performance assessment (from Fig. 4b it seems 6.3 microamps because probably a different assumption has been made on the gate workfunctions)”*

Answer 3: We have indeed adjusted the values of the workfunction to match the benchmarking.

Question 4. *“Add a Figure similar to Fig. 4b for the large-gap TIFET at $V_{\text{ds}} = 0.3\text{ V}$, so that one can appreciate the I_{off} .”*

Answer 4: We have added a Fig. 5b similar to Fig. 4b for the large gap TI FET and mentioned the off-current in the caption

Question 5. *“Comment the issue of the static power consumption (with quantitative figures) when discussing performance metrics.”*

Answer 5: We have included a paragraph “In the methodology from Ref. 46, 47, off-current is not considered. Given the significant off-current that can be observed in Fig. 4b (I_{off} 0.02 μA (0.09 μA), the TI FET would have a large static power $P_{\text{static}} = V_{\text{dd}}I_{\text{off}} = 4$ nW (27 nW) in a conventional CMOS setting. Active power consumption at a switching speed of $1/t_{\text{int}} = 0.6$ THz (1.1 THz) and assuming an activity factor of 1 would lead to $P_{\text{active}} = E_{\text{ic}}/t_{\text{ic}} = 0.8$ μW (3.9 μW) where active power dominates over static power by a factor by 200 (144). In a practical setting, however, an activity factor of 1 is not obtainable and circuit switching speed must be slower than $1/t_{\text{int}}$. Switching more slowly than 1 GHz or at lower activity factors, static power consumption would inevitably come to dominate compromising energy efficiency. Static power consumption can be reduced to some extent by changing the workfunction to reduce the off-current significantly and the on-current less significantly. Nevertheless, the off-current will always be large compared to low-standby power CMOS and the main target TI FET logic applications would thus be found in the realm of high-performance operation where high activity factors can be maintained.”

Reviewer #3 wrote:

“I’m satisfied with how the authors have addressed the criticisms by me and the other reviewers, I thus support the publication of the present manuscript in Nature Communications.”

Response: We thank the reviewer again for his/her comments.